# WeFT: Weighted Entropy-driven Fine-Tuning for dLLMs

## Abstract

Diffusion models have recently shown strong potential in language modeling, offering faster generation compared to traditional autoregressive approaches. However, applying supervised fine-tuning (SFT) to diffusion models remains challenging, as they lack precise probability estimates at each denoising step. While the diffusion mechanism enables the model to reason over entire sequences, it also makes the generation process less predictable and often inconsistent. This highlights the importance of controlling key tokens that guide the direction of generation. To address this issue, we propose WeFT, a weighted SFT method for diffusion language models, where tokens are assigned different weights based on their entropy. Derived from diffusion theory, WeFT delivers substantial gains: training on s1K, s1K-1.1, and 3k samples from open-r1, it achieves relative improvements of 39%, 64%, and 83% over standard SFT on four widely used reasoning benchmarks (Sudoku, Countdown, GSM8K, and MATH-500). The code is provided in the supplementary material.

## 1 Introduction

Recently, the landscape of large language modeling has been reshaped by the emergence of diffusion-based approaches (Nie et al., 2025; Ou et al., 2025; Yang et al., 2025), which offer a fundamentally different perspective on sequence generation compared to autoregressive (AR) methods Radford et al. (2018; 2019); Brown et al. (2020). Rather than relying on strictly sequential decoding, diffusion large language models (dLLMs) employ iterative refinement procedures that enable efficient parallel generation and open new possibilities for scaling language models.

Currently, similar to AR models, the training paradigm of dLLMs is generally divided into three stages: pre-training, instruction tuning, and reinforcement learning (RL). Among them, the pre-training and instruction tuning stages adopt an SFT loss specifically derived for dLLMs (Nie et al., 2025), while the RL stage employs a loss estimated on the basis of the PPO algorithm (Schulman et al., 2017) and tailored to the characteristics of dLLMs (Zhu et al., 2025).

Specifically, when designing the SFT objective, (Nie et al., 2025) model the generation process of dLLMs as a continuous-time diffusion process, where the diffusion timestep $t$ ranges within $[0, 1]$. Here, $t = 0$ denotes a completely original, non-masked sequence, and $t = 1$ denotes a fully masked sequence. As illustrated in Figure 1(a), let the original response of the model be $x_0$. At a given timestep $t$, each token is masked with probability $t$, resulting in a partially masked response $x_t$. The training objective of dLLMs is then to enable the model to reconstruct the masked parts of $x_0$ given the partially masked input $x_t$. Based on this formulation, the loss is defined as follows:

$$L = \sum_i \mathbb{E}_t \left[ \mathbf{1}[x_t^i = \mathbf{M}] \frac{1}{t} \log \left( x_0^i \mid x_t \right) \right] \tag{1}$$

This design of the SFT loss is intuitive and aligns well with the underlying rationale of diffusion theory. However, it presents several critical limitations. Among them, one of the most prominent issues is that the loss implicitly assumes a uniform masking rate across all tokens, thereby treating each token as equally important throughout the diffusion process. This assumption overlooks the inherent heterogeneity in token significance: tokens that play a central role in planning and reasoning should arguably receive greater emphasis during training than other tokens.

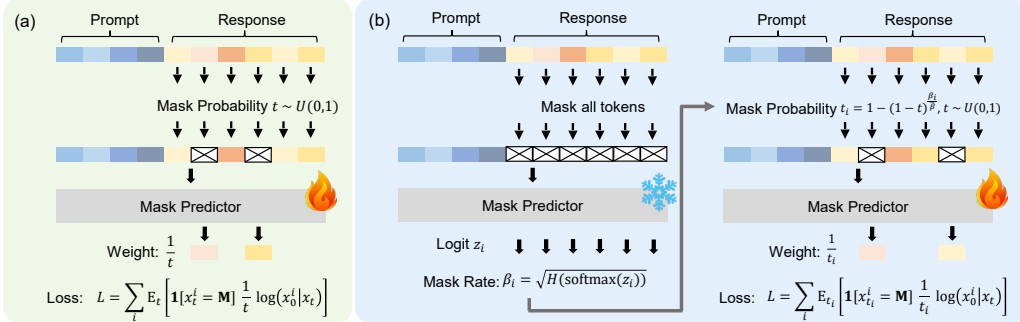

Figure 1: (a) The SFT pipeline. A timestep $t$ is uniformly sampled from $[0, 1]$, and each token is masked independently with probability $t$. The training objective is to predict the masked tokens accurately based on the unmasked ones. (b) The WeFT pipeline. In each training step, we perform two forward passes. During the first forward pass, we mask the entire answer and estimate the masking rate $\beta_i$ for each token by computing its predictive entropy. In the second forward pass, the $i$-th token is masked with probability $t_i$ (computed from $\beta_i$), and its training weight is set to $\frac{1}{t_i}$. Tokens with higher entropy are more likely to be masked and thus receive stronger training signals.

To address this discrepancy, we introduce WeFT, a token-aware weighting mechanism based on predictive uncertainty. Specifically, we find that the entropy of the model's predictive distribution over tokens serves as a reliable metric for token importance at the current training stage. Tokens associated with higher entropy are typically those that relate to reasoning or planning, and they also correspond to positions where the model exhibits greater uncertainty in generation. By prioritizing these high-entropy tokens during training via an adaptive weighting strategy applied to the SFT loss, we can encourage the model to allocate more capacity to the parts of the sequence that are crucial for planning and reasoning.

Based on the fundamental principles of diffusion models, we derive a weighted SFT loss formulation that is theoretically consistent with diffusion dynamics. Empirically, we evaluate this approach on four widely used benchmarks, Sudoku (Arel, 2025), Countdown (Pan et al., 2025), GSM8K (Cobbe et al., 2021), and MATH500 (Lightman et al., 2024), by comparing models fine-tuned with the standard SFT algorithm against those fine-tuned with WeFT under identical training data. Specifically, our method is trained on s1K, s1K-1.1 (Muennighoff et al., 2025), and 3k samples drawn from the Mixture-of-Thoughts dataset of open-r1 (HuggingFace, 2025). Across the four reasoning benchmarks, our approach achieves relative improvements of 39%, 64%, and 83% over standard SFT.

Moreover, we observe that these gains persist through subsequent RL training stages. Compared with models cold-started using SFT, those initialized with WeFT achieve a relative performance improvement of 49% in later RL training. In addition, ablation studies further confirm that both the theoretically derived weighted SFT loss and the entropy-based weighting scheme are indispensable for achieving the observed performance improvements. Finally, the time efficiency analysis shows that WeFT introduces only minimal computational overhead, with merely a 24% increase in training time compared to SFT.

In summary, our core contributions are as follows:

- We derive a theoretically grounded formulation of the weighted SFT loss from the perspective of diffusion processes, ensuring consistency with the underlying principles of diffusion-based generation.

- We identify token-level predictive entropy as a reasonable and effective metric for capturing token importance, and we argue that tokens with higher entropy should be trained more frequently to enhance the model's ability to handle planning and reasoning.

- Building upon these insights, we propose WeFT, a weighted SFT algorithm that consistently outperforms the standard SFT approach across multiple training sets and benchmarks. Ablation studies further validate the necessity and effectiveness of each design choice in WeFT.

## 2 PRELIMINARIES: CONTINUOUS-TIME DISCRETE DIFFUSION MODEL

Following (Ou et al., 2025), we model dLLMs as a *Continuous-Time Discrete Diffusion Model* (Sun et al., 2023). Intuitively, in each infinitesimal time interval $\Delta t$, every token has a certain probability of being masked.

Formally, this process can be described as a time-dependent Markov chain, whose transition dynamics are governed by a matrix $Q_t$. For a transition from state $x$ to $y$ within $(t, t + \Delta t)$, we have

$$p_{t+\Delta t|t}(y|x) = \begin{cases} Q_t(x, y)\Delta t + o(t) & \text{if } x \neq y \\ 1 + Q_t(x, x)\Delta t + o(t) & \text{if } x = y \end{cases} \tag{2}$$

where $Q_t(x, y) \geq 0$ for $x \neq y$ and $Q_t(x, x) < 0$. Thus, $Q_t(x, y)$ specifies the instantaneous transition rate from state $x$ to $y$.

In practice, it is common to assume a *time–factorized form* (Campbell et al., 2022), namely

$$Q_t = f(t)Q, \tag{3}$$

where $Q$ is a constant matrix and $f(t)$ is a scalar function controlling the evolution speed.

Let $P_{t|s}(x, y) = p_{t|s}(y|x)$ denote the transition probability matrix from time $s$ to $t$. Then, $P_{t|s}$ satisfies the Kolmogorov's forward equation (Anderson, 2012):

$$\frac{d}{dt}P_{t|s} = f(t)P_{t|s}Q. \tag{4}$$

The solution is given by

$$P_{t|s} = \exp\left(Q \int_s^t f(u)\,du\right), \tag{5}$$

where $\exp$ denotes the matrix exponential.

For notational convenience, we define

$$\bar{f}(x) = \int_0^x f(t)\,dt. \tag{6}$$

With this shorthand, the solution can be equivalently written as

$$P_{t|s} = \exp\big((\bar{f}(t) - \bar{f}(s))Q\big). \tag{7}$$

Intuitively, $s$ corresponds to the sequence before masking at the current step, $t$ corresponds to the sequence after masking at the current step, and $Q$ represents the masking rate of each token. In prior work, it is commonly assumed that the $Q$ matrix takes the following form: (Ou et al., 2025; Nie et al., 2025)

$$Q = \begin{bmatrix} -1 & 0 & \cdots & 0 & 1 \\ 0 & -1 & \cdots & 0 & 1 \\ \vdots & \vdots & \ddots & \vdots & \vdots \\ 0 & 0 & \cdots & -1 & 1 \\ 0 & 0 & \cdots & 0 & 0 \end{bmatrix} \tag{8}$$

Under this assumption, all tokens gradually diffuse into an absorbing state [M], that is, they become masked.

## 3 METHOD

### 3.1 OVERVIEW

We propose WeFT (Weighted Entropy-driven Fine-Tuning), a weighted SFT method designed to focus training on tokens that carry more information. In practice, we quantify token importance using the square root of token entropy.

As shown in Figure 1(a), the original SFT approach masks all tokens with the same probability $t$, and each token contributes equally to the loss function with a weight of $\frac{1}{t}$. In contrast, Figure 1(b) illustrates the WeFT pipeline. In each training step, we perform two forward passes. In the first pass, the entire answer is masked, and the masking rate $\beta_i$ for each token is estimated based on its predictive entropy. In the second pass, each token is independently masked with probability $t_i$ (computed from $\beta_i$), and its training weight is assigned as $\frac{1}{t_i}$. This design ensures that tokens with higher entropy are more likely to be masked and trained more frequently, thereby improving the model's ability to capture complex dependencies.

The derivation of the loss function is presented in Section 3.2, while the details of the weighting scheme are provided in Section 3.3. A detailed description of the implementation of WeFT can be found in Appendix C.

### 3.2 WEIGHTED SFT LOSS FUNCTION DERIVATION

As noted earlier, prior work typically assumes that all tokens are masked at the same rate, leading to a $Q$ matrix that takes the form of an absorbing matrix composed of 1 and $-1$. In contrast, we assign each token a distinct masking rate $\beta$, from which we derive a weighted SFT formulation. Importantly, the resulting SFT loss remains consistent with the fundamental properties of diffusion, since the $Q$ matrix is still well-defined under this construction. Namely, we now set $Q$ as:

$$
Q = \begin{bmatrix}
-\beta_1 & 0 & \cdots & 0 & \beta_1 \\
0 & -\beta_2 & \cdots & 0 & \beta_2 \\
\vdots & \vdots & \ddots & \vdots & \vdots \\
0 & 0 & \cdots & -\beta_{n-1} & \beta_{n-1} \\
0 & 0 & \cdots & 0 & 0
\end{bmatrix}
\tag{9}
$$

Based on the newly defined form of the $Q$ matrix, and following the derivation in (Ou et al., 2025), we obtain the modified formulation of the SFT loss. The detailed derivation is provided in Appendix A.

**Theorem 1.** *Assuming the $Q$ matrix takes the form given in Equation 9, let the initial sequence be $x_0$ and the sequence at time $t$ be $x_t$. Under this setting, the $i$-th token is masked with probability $t_i = 1 - (1-t)^{\frac{\beta_{x^i}}{\beta_{ref}}}$, where $\beta_{x^i}$ denotes the masking rate of the $i$-th token, and $\beta_{ref}$ is a specified reference masking rate. Moreover, the Weighted SFT loss can be derived as follows:*

$$
L = \sum_i \mathbb{E}_{t_i} \left[ \mathbf{1}\left[ x_{t_i}^i = \mathbf{M} \right] \frac{1}{t_i} \log \left( x_0^i \mid x_t \right) \right]
\tag{10}
$$

Intuitively, under the weighted SFT loss, a token $x_i$ with a larger masking rate $\beta_{x^i}$ corresponds to a larger $t_i$, making it more likely to be masked and subsequently learned. Here, $\beta_{\text{ref}}$ can be any reference value. Empirically, we set $\beta_{\text{ref}}$ to the mean of all $\beta$ values to enhance numerical stability.

### 3.3 MOTIVATION OF USING ENTROPY AS THE MASKING RATE METRIC

Having established the form of the weighted SFT loss, we next investigate how to assign masking rate to tokens. First, we require that the chosen metric does not introduce excessive computational overhead. In particular, it is infeasible to compute such a metric for every token individually, and we can only afford $O(1)$ additional forward passes in total. For this reason, we consider masking all the answers and then examining the distribution of the model's output logits $z_i$ to be a practical choice. Specifically, given a problem $x$ and its answer $y$, we mask the entire answer and compute

$$
z_i = \text{model}(\cdot \mid x, [M]).\text{logits}.
\tag{11}
$$

We then determine the masking rate $\beta_i$ of the $i$-th token. Based on our analysis, the entropy of the distribution of $z_i$ serves as an effective metric. As shown in Figure 2, we visualize the entropy distribution among the 100 most frequent tokens during training, contrasting tokens with higher versus lower entropy. It can be observed that tokens with higher entropy tend to carry richer information

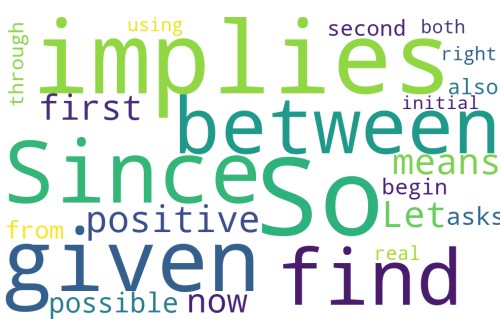 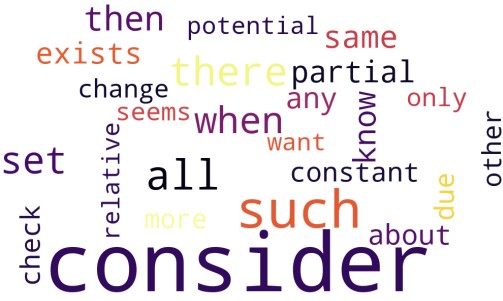

| Frequent tokens with high entropy | Frequent tokens with low entropy |

Figure 2: Visualization of high-frequency tokens with different entropy levels. Tokens with higher entropy tend to carry more information and play a crucial role in shaping the structure of generated responses.

and play a more critical role in shaping the structure of generated responses. For example, words such as "first" and "second" exhibit higher entropy, often signaling logical or sequential structures, whereas words like "all" and "such" show lower entropy and function primarily as auxiliary words, contributing less to semantic content.

Accordingly, we define the masking rate $\beta_i$ as

$$\beta_i = H(\text{softmax}(z_i)). \tag{12}$$

However, we empirically find that the raw entropy values exhibit considerable variance, which may destabilize training. To mitigate this, we instead use the square root of entropy, which yields more stable optimization in practice. This design choice is further validated by experiments in Section 4.4.3.

$$\beta_i = \sqrt{H(\text{softmax}(z_i))}. \tag{13}$$

## 4 EXPERIMENT

In this section, we first describe the experimental setup, followed by a comparison of the performance of WeFT and the standard SFT algorithm across multiple training datasets and benchmarks. We then analyze the performance of models trained with WeFT during subsequent RL training, conduct ablation studies to validate the necessity of each design component in WeFT, and finally provide a runtime comparison between WeFT and SFT under identical settings.

### 4.1 EXPERIMENTAL SETUP

We adopt LLaDA-8B-Instruct (Nie et al., 2025) as the base model and apply LoRA (Hu et al., 2022) for fine-tuning. For the training datasets, we use s1K, s1K-1.1, and openr1-3k to evaluate the performance of WeFT and SFT. Among them, s1K and s1K-1.1 are reasoning datasets proposed by (Muennighoff et al., 2025), while openr1-3k is constructed by sampling 3k examples from the Mixture-of-Thoughts dataset of open-r1 (HuggingFace, 2025), with sequence lengths less than 2.5k under the open-r1 encoder. The evaluation is conducted on the following benchmarks:

- **Sudoku** (Arel, 2025): A symbolic reasoning benchmark requiring models to solve Sudoku puzzles through step-by-step logical deduction.
- **Countdown** (Pan et al., 2025): A numerical reasoning dataset where models must combine given numbers using arithmetic operations to reach a target value.
- **GSM8K** (Cobbe et al., 2021): A widely used benchmark of grade-school math word problems designed to evaluate arithmetic and reasoning abilities.
- **MATH500** (Lightman et al., 2024): A subset of the MATH dataset containing 500 challenging competition-level math problems that test advanced mathematical reasoning.

The detailed training and evaluation hyperparameters are provided in Appendix B.

## 4.2 MAIN RESULTS

As shown in Table 1, we compare the performance of WeFT and the standard SFT method across different training datasets and benchmarks. Compared with the base LLaDA-8B-Instruct model, both SFT and WeFT improve performance on all tasks. Notably, WeFT consistently achieves higher accuracy than SFT, with a 39% improvement on the s1K dataset, a 64% improvement on the s1K-1.1 dataset, and a 83% improvement on the openr1-3k dataset, demonstrating that our method effectively leverages token importance and structured information. These results indicate that WeFT provides robust and reliable performance gains across tasks and datasets, particularly for tasks requiring complex reasoning.

| Model | Sudoku 0-shot | Countdown 0-shot | GSM8K 0-shot | MATH500 0-shot | Average |
|---|---|---|---|---|---|
| **Base Model** | | | | | |
| LLaDA-8B-Instruct | 5.5 | 16.0 | 76.7 | 32.4 | 32.6 |
| **s1K Dataset** | | | | | |
| + SFT | 4.6 (-0.9) | 23.8 (+7.8) | **78.8 (+2.1)** | 32.6 (+0.2) | 34.9 (+2.3) |
| + WeFT | **7.8 (+2.3)** | **24.6 (+8.6)** | 78.0 (+1.3) | **33.0 (+0.6)** | **35.8 (+3.2)** |
| **s1K-1.1 Dataset** | | | | | |
| + SFT | 4.2 (-1.3) | **21.5 (+5.5)** | **78.4 (+1.7)** | 30.8 (-1.6) | 33.7 (+1.1) |
| + WeFT | **6.8 (+1.3)** | 20.3 (+4.3) | 78.1 (+1.4) | **32.6 (+0.2)** | **34.4 (+1.8)** |
| **openr1-3k Dataset** | | | | | |
| + SFT | 6.3 (+0.8) | 17.2 (+1.2) | 77.3 (+0.6) | 32.0 (-0.4) | 33.2 (+0.6) |
| + WeFT | **6.9 (+1.4)** | **17.6 (+1.6)** | **77.5 (+0.8)** | **32.8 (+0.4)** | **33.7 (+1.1)** |

Table 1: Comparison of the performance of WeFT and SFT across different training datasets and benchmarks. The table shows that WeFT achieves substantial percentage improvements over SFT.

## 4.3 SFT ROLE IN SUBSEQUENT RL

Reinforcement learning (RL) post-training has become a crucial stage in the training of large language models (DeepSeek-AI, 2025). Supervised fine-tuning (SFT) cold-start plays an important role in preparing the model for subsequent RL, as SFT establishes a reliable initial policy which can then be further refined by RL objectives. Therefore, it is essential to verify whether models trained with WeFT can maintain their advantages during subsequent RL training. To this end, we fine-tune models on the s1K dataset using both WeFT and SFT, and then further train them with RL on the Sudoku dataset for comparison.

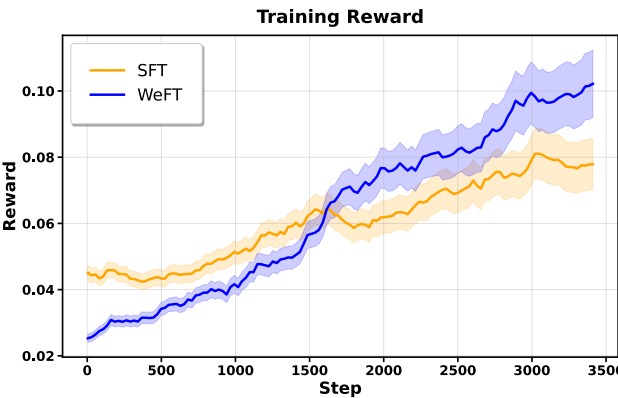

Figure 3: Reward curves of models cold-started with WeFT and SFT during subsequent reinforcement learning training. The curves are smoothed using a time-weighted EMA. As shown, the model initialized with WeFT achieves higher rewards.

As shown in Figure 3, models initialized with WeFT achieve higher rewards throughout RL training. Moreover, as reported in Table 2, the WeFT-trained model achieves significantly better evaluation results, with a 49% relative performance gain. These findings demonstrate that models cold-started with WeFTare highly effective in guiding subsequent RL learning.

| Model | LLaDA-8B-Instruct | + SFT | + WeFT | + SFT + RL | +WeFT + RL |
|---|---|---|---|---|---|
| Sudoku | 5.5 | 4.6 | **7.8** | 8.9 | **13.3** |

Table 2: Evaluation results of models cold-started with WeFT and SFT during subsequent reinforcement learning training. As shown in the table, the model initialized with WeFT achieves significantly higher evaluation accuracy.

## 4.4 ABLATION STUDY

In this subsection, we conduct ablation studies to validate the design choices of WeFT, focusing on three aspects: the necessity of using entropy as the metric, the validity of the theoretically derived loss function, and the empirical necessity of applying the square root of entropy during training.

### 4.4.1 THE NECESSITY OF USING ENTROPY AS THE METRIC

To demonstrate the necessity of entropy-based weighting, we compared our method with a baseline using the negative log-likelihood (NLL). Unlike entropy-based weighting, NLL emphasizes tokens that appear less frequently rather than those that are difficult to predict or carry more information. Assuming the logits are $z_i$, the masking rate $\beta_i$ for the $i$-th token under the NLL setting is:

$$\beta_i = -\log\big(\mathrm{softmax}(z_i)^{x_0^i}\big) \tag{14}$$

| Model | Sudoku 0-shot | Countdown 0-shot | GSM8K 0-shot | MATH500 0-shot | Average |
|---|---|---|---|---|---|
| **Base Model** | | | | | |
| LLaDA-8B-Instruct | 5.5 | 16.0 | 76.7 | 32.4 | 32.6 |
| **s1K Dataset** | | | | | |
| + SFT | 4.6 (-0.9) | 23.8 (+7.8) | 78.8 (+2.1) | 32.6 (+0.2) | 34.9 (+2.3) |
| + Weighted SFT (- log p) | 3.0 (-2.5) | 16.4 (+0.4) | 75.4 (-1.3) | 29.2 (-3.2) | 31.0 (-1.6) |
| + WeFT | 7.8 (+2.3) | 24.6 (+8.6) | 78.0 (+1.3) | 33.0 (+0.6) | 35.8 (+3.2) |

Table 3: Comparison between entropy-based weighting and NLL-based weighting. As shown in the table, the SFT algorithm with NLL-based weighting performs significantly worse than the entropy-based variant, and on many tasks even underperforms the original model.

As shown in Table 3, the SFT method weighted by NLL performs poorly, in some cases even worse than the original base model. This underscores the importance of focusing on high-entropy, informative tokens for achieving effective fine-tuning performance.

### 4.4.2 THE VALIDITY OF THE THEORETICALLY DERIVED LOSS FUNCTION

Our loss function is derived from the continuous-time discrete diffusion model combined with the Denoising Score Entropy (DSE) loss. As a result, our model preserves the inherent characteristics of the diffusion process more effectively than methods that simply add a weight to the loss function:

$$L = \sum_i \mathbb{E}_t \left[ \mathbf{1}\big[x_t^i = \mathbf{M}\big] \frac{w_i}{t} \log\big(x_0^i \mid x_t\big) \right] \tag{15}$$

To illustrate this, we compare our approach with two weighted SFT methods by applying a simple token-wise weighting, namely using our square-root entropy as the weight, and using the Dream (Ye et al., 2025) method. Specifically, when using our square-root entropy, the weight of the $i$-th token is:

$$w_i = \sqrt{H(\mathrm{softmax}(z_i))} \tag{16}$$

When using the weighting method from Dream, the weight of the $i$-th token is:

$$w_i = \frac{1}{2} \sum_{j=1}^{N} \mathbf{1}\left[x_t^j \neq \mathbf{M}\right] \mathrm{Geo}(p, |j - i| - 1) \tag{17}$$

Here, Geo denotes the geometric distribution, and $p$ controls the sharpness of the distribution. Since the original work did not provide the corresponding parameter setting, we set it to a moderate value of 0.3.

Table 4 presents a comparison between WeFT, WeFT (SW) (i.e., simple token-wise weighting using our square-root entropy as the weight), and Dream. Dream yields only minimal improvements and, in some cases, performs worse than standard SFT. WeFT (SW) provides some gains but still performs worse than SFT, whereas WeFT significantly outperforms SFT. This demonstrates that the diffusion loss derived from theoretical principles possesses superior properties.

| Model | Sudoku 0-shot | Countdown 0-shot | GSM8K 0-shot | MATH500 0-shot | Average |
|---|---|---|---|---|---|
| **Base Model** | | | | | |
| LLaDA-8B-Instruct | 5.5 | 16.0 | 76.7 | 32.4 | 32.6 |
| **s1K Dataset** | | | | | |
| + SFT | 4.6 (-0.9) | 23.8 (+7.8) | 78.8 (+2.1) | 32.6 (+0.2) | 34.9 (+2.3) |
| + Dream | 3.3 (-2.2) | 21.1 (+5.1) | 77.0 (+0.3) | 30.0 (-2.4) | 32.8 (+0.2) |
| + WeFT (SW) | 5.3 (-0.2) | 23.8 (+7.8) | 75.9 (-0.8) | 31.6 (-0.8) | 34.2 (+1.6) |
| + WeFT | 7.8 (+2.3) | 24.6 (+8.6) | 78.0 (+1.3) | 33.0 (+0.6) | 35.8 (+3.2) |

Table 4: Comparison between WeFT, WeFT (SW) (simple token-wise weighting using our square-root entropy as the weight), and Dream. Dream yields negligible improvements. WeFT (SW) provides some gains, but still performs worse than SFT, whereas WeFT significantly outperforms SFT.

### 4.4.3 THE EMPIRICAL NECESSITY OF APPLYING THE SQUARE ROOT OF ENTROPY DURING TRAINING

Below, we explain why we choose the square root of entropy rather than entropy as the metric. As shown in Table 5, we observed that directly using token weights based on raw entropy leads to large variations across tokens, which can significantly reduce training stability and even cause oscillations. To address this issue, we implement WeFT using the square root of entropy as the weighting factor. This modification effectively reduces the scale of the gradient norms, resulting in more stable training and improved overall convergence behavior.

| Gradient Norm | Entropy | $\sqrt{\text{Entropy}}$ |
|---|---|---|
| Max | 646.1 | 46.2 |
| Median | 0.47 | 0.43 |
| Average | 2.02 | 1.03 |

Table 5: Comparison of gradient norms during training when using entropy versus square root entropy. Using entropy sometimes leads to extremely large gradient norms, resulting in instability, whereas using square root entropy effectively alleviates this issue.

## 4.5 TIME EFFICIENCY ANALYSIS

Compared to SFT, WeFT introduces one additional forward pass per training step. To evaluate the computational overhead, we measured the time required to train both SFT and WeFT for 20 epochs on the s1K dataset using 4 H100 GPUs. As shown in Figure 4, the results show that WeFT incurs only a modest increase in training time (approximately 24% more than standard SFT), demonstrating that the performance gains of WeFT come with minimal additional computational cost.

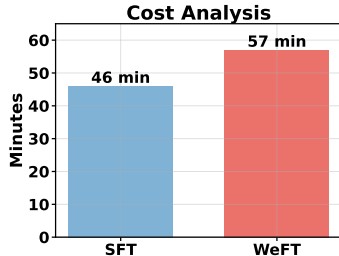

Figure 4: Time efficiency analysis of SFT and WeFT.

## 5 RELATED WORK

### 5.1 DIFFUSION LARGE LANGUAGE MODELS

Diffusion models, initially developed for continuous data, have recently been adapted to discrete sequences, including natural language. Early work such as Structured Denoising Diffusion Models (D3PMs) (Austin et al., 2021) extended discrete diffusion to text, showing competitive performance in character-level generation and large vocabulary language modeling. DiffusionBERT (He et al., 2023) combined masked language modeling with diffusion, improving generative quality over conventional masked LMs. Later, masked diffusion frameworks (Sahoo et al., 2024; Shi et al., 2024) introduced simplified and generalized training objectives that reduce complexity while maintaining performance, and support state-dependent masking strategies. Diffusion Large Language Models (dLLMs) including LLaDA (Nie et al., 2025) and Dream (Ye et al., 2025) demonstrated that diffusion-based models can scale to billions of parameters, achieving strong context learning and instruction-following capabilities. Techniques such as variance-reduced preference optimization (Zhu et al., 2025) and reinforcement of lateral thought chains (Huang et al., 2025) further enhance generative stability and reasoning ability, while efficient perplexity bounds and ratio matching (Haxholli et al., 2025) improve sampling efficiency. More recently, works such as d1 (Zhao et al., 2025) and wd1 (Tang et al., 2025) have explored the possibility of applying reinforcement learning on top of existing dLLMs.

### 5.2 WEIGHTED METHODS IN LANGUAGE MODEL LEARNING

A complementary line of research focuses on improving large language models through weighted supervised fine-tuning (SFT) and reinforcement learning. Weighted SFT techniques aim to prioritize informative tokens. DFT (Wu et al., 2025) addresses the limited generalization of SFT by dynamically rescaling token-level objectives to stabilize gradient updates. Rho-1 (Lin et al., 2024) challenges the conventional uniform next-token prediction objective by selectively training on useful tokens identified via a reference model. Similarly, Luo et al. (Luo et al., 2023) propose a reweighting strategy for active learning in NER, assigning dynamic smoothed weights to tokens to boost performance. In reinforcement learning, applying weighting to improve learning effectiveness is also a common strategy. (Cui et al., 2025) investigate the entropy dynamics of reinforcement learning for reasoning LLMs, and propose entropy-based weighting methods to prevent entropy collapse and encourage exploration. (Wang et al., 2025) highlight the critical role of high-entropy minority tokens in RLVR, demonstrating that restricting updates to these tokens significantly enhances reasoning capabilities. Together, these approaches demonstrate that token-level weighting is an important and effective way to improve the optimization of language models.

## 6 CONCLUSION

We introduced WeFT, a theoretically grounded weighted SFT algorithm for diffusion-based language models. By leveraging token-level entropy as a measure of importance, WeFT prioritizes high-uncertainty tokens and preserves the diffusion process properties. Experiments across multiple benchmarks show substantial and consistent improvements over standard SFT, with gains persisting into RL training, all at minimal computational overhead. These results highlight entropy-aware weighting as a simple yet powerful strategy for enhancing reasoning and stability in diffusion-based LLMs.

ETHICS STATEMENT

This paper aims to advance the field of machine learning. The work is solely centered on the methodology itself; how it is applied and for what purpose is entirely up to the users. As with other large-model research, models trained with our algorithm may generate content that is factually incorrect or potentially harmful, and such outputs should be carefully vetted by users.

REPRODUCIBILITY STATEMENT

We provide a detailed description of the implementation of WeFT in Appendix C, and the specific hyperparameters used for training and evaluation in Appendix B. Based on the algorithm implementation in Appendix C, WeFT can be realized with only minor modifications to existing training frameworks for dLLMs, while the evaluation can be directly conducted using existing evaluation code. In addition, the code is provided in the supplementary material.

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

# A PROOF OF THE WEIGHTED SFT LOSS

In order to prove this theorem, we will first prove two lemmas, and then proceed to prove the main theorem. Our proof follows that of (Ou et al., 2025), with the key difference that our $Q$ matrix incorporates varying masking rates $\beta$, whereas (Ou et al., 2025) assumes all masking rates are identical.

## A.1 PROOF OF LEMMAS

Let the original response of the model be $x_0$. At a given timestep $t$, let the partially masked response be $x_t$, and $[M]$ denotes the mask. $\beta_{x_t^i}$ denotes the masking rate corresponding to the $i$-th token. The definition of $\bar{f}$ can be found in Equation 6.

**Lemma 1.** *Let $x_t^i = [M]$ and $x'^i_t \neq [M]$. Then we have*

$$\frac{p_t(x_t^1, \ldots, x'^i_t, \ldots, x_t^d)}{p_t(x_t^1, \ldots, x_t^i, \ldots, x_t^d)} = \frac{\exp(-\beta_{x_t^i} \, \bar{f}(t))}{1 - \exp(-\beta_{x_t^i} \, \bar{f}(t))} \, p_0(x'^i_t \mid x_t^{\text{unmasked}}). \tag{18}$$

*On the other hand, if $x_t^i \neq [M]$, then*

$$\frac{p_t(x_t^1, \ldots, x'^i_t, \ldots, x_t^d)}{p_t(x_t^1, \ldots, x_t^i, \ldots, x_t^d)} = 0. \tag{19}$$

Intuitively, this theorem characterizes the exact transition probabilities during the training process. It also establishes a connection between the concrete score function and the clean data distribution via a scalar factor.

*Proof.* We first calculate $p_{t|0}(x_t|x_0)$:

Based on the definition of $Q_t$, we can easily see that $p_{t|0}(x_t|x_0) = 0$ if $x_t \neq x_0$ and $x_t \neq [M]$

Now we divide the time interval $(0, t)$ into $n$ intervals, namely $(0, s_1), (s_1, s_2), \cdots, (s_{n-1}, t)$. Here we denote $s_0 = 0, s_n = t$

then the total probability of $p_{t|0}(x_t = x_0|x_0)$ is

$$\prod_{i=0}^{n-1} p_{s_{i+1}|s_i}(x_{s_{i+1}} = x_{s_i}|x_{s_i})$$

$$= \prod_{i=0}^{n-1} (1 + Q_{s_i}(x_{s_i}, x_{s_i})(s_{i+1} - s_i) + o(s_{i+1} - s_i))$$

$$= \exp(\sum_{i=0}^{n-1} \ln(1 + Q_{s_i}(x_{s_i}, x_{s_i})(s_{i+1} - s_i) + o(s_{i+1} - s_i)))$$

$$= \exp(\sum_{i=0}^{n-1} [f(s_i)Q(x_{s_i}, x_{s_i})(s_{i+1} - s_i) + o(s_{i+1} - s_i)])$$

Now we select the the appropriate $n$ and $s_1, \ldots, s_{n-1}$ such that $n \to +\infty$ and $\max(s_{i+1} - s_i) \to 0$, recalling the definition of integral, we obtain:

$$\exp(\sum_{i=0}^{n-1} [f(s_i)Q(x_{s_i}, x_{s_i})(s_{i+1} - s_i) + o(s_{i+1} - s_i)]) \to \exp(-\beta_{x_0}\bar{f}(t))$$

Thus, we have

$$p_{t|0}(x_t|x_0) = \begin{cases} \exp(-\beta_{x_0}\bar{f}(t)) & \text{if } x_t = x_0 \\ 1 - \exp(-\beta_{x_0}\bar{f}(t)) & \text{if } x_t = [M] \\ 0 & \text{otherwise} \end{cases}$$

Now, based on the above result, recall the independence of different dimensions in the diffusion process, we can get a stronger result:

**Lemma 2.** *Suppose we have $x_t$ in which the tokens $a_1, a_2, ..., a_k$ are masked and $b_1, b_2, ..., b_{n-k}$ are unmasked, then:*

$$p_t(x_t) = \prod_{i=1}^{k}(1 - \exp(-\beta_{a_i}\bar{f}(t)))\prod_{j=1}^{n-k}\exp(-\beta_{b_j}\bar{f}(t))p_0(x_t^{unmasked}) \tag{20}$$

*Proof.* Exploiting the independence across dimensions in the diffusion process, we may, without loss of generality, set $a_i = i$ and $b_i = k + i$. Equivalently, this corresponds to masking the first $k$ tokens while leaving the remaining $n - k$ tokens unmasked.

Then we obtain

$$p_t([M][M]\dots[M]x_t^{k+1}x_t^{k+2}\dots x_t^n)$$

$$= \sum_{x_0} p_{t|0}([M][M]\dots[M]x_t^{k+1}x_t^{k+2}\dots x_t^n|x_0)p_0(x_0)$$

$$= \sum_{x_0^1\dots x_0^n} p_{t|0}([M][M]\dots[M]x_t^{k+1}x_t^{k+2}\dots x_t^n|x_0^1, x_0^2, \dots x_0^n)p_0(x_0^1, \dots x_0^n)$$

$$= \sum_{x_0^1\dots x_0^n} \prod_{i=1}^{k}p_{t|0}^k([M]|x_0^k) \prod_{i=k+1}^{n} p_{t|0}(x_t^k|x_0^k)p_0(x_0^1, \dots x_0^n)$$

$$= \sum_{x_0^1\dots x_0^n} \prod_{i=1}^{k}(1 - \exp(-\beta_i\bar{f}(t))) \prod_{i=k+1}^{n} \exp(-\beta_i\bar{f}(t))p_0(x_0^1, \dots x_0^n)$$

$$= \sum_{x_0^{k+1}\dots x_0^n} \prod_{i=1}^{k}(1 - \exp(-\beta_i\bar{f}(t))) \prod_{i=k+1}^{n} \exp(-\beta_i\bar{f}(t))p_0(x_0^{k+1}, \dots x_0^n)$$

Since $p_0(x_0^{k+1}, \dots, x_0^n)$ can be expressed as $p_0(x_t^{\text{unmasked}})$, the desired result follows.

$\square$

Now we append $x_t'$ and $x_t$ to the above result and we obtain the result of Lemma 1.

$\square$

### A.2 PROOF OF THE MAIN THEOREM

**Theorem 1.** *Assuming the Q matrix takes the form given in Equation 9, let the initial sequence be $x_0$ and the sequence at time $t$ be $x_t$. Under this setting, the $i$-th token is masked with probability $t_i = 1 - (1 - t)^{\frac{\beta_{x^i}}{\beta_{ref}}}$, where $\beta_{x^i}$ denotes the masking rate of the $i$-th token, and $\beta_{ref}$ is a specified reference masking rate. Moreover, the Weighted SFT loss can be derived as follows:*

$$L = \sum_i \mathbb{E}_{t_i}\left[\mathbf{1}\left[x_{t_i}^i = \mathbf{M}\right]\frac{1}{t_i}\log\left(x_0^i \mid x_t\right)\right] \tag{10}$$

*Proof.* Now we consider the Denoising Score Entropy, introduced in Lou et al. (2024). DSE loss provides a principled training objective for diffusion models by measuring the discrepancy between the model score function and the true score of the perturbed data distribution.

The DSE loss can be written as:

$$\int_0^T \mathbb{E}_{x_t \sim p_{t|0}(x_t|x_0)} \sum_{x_t' \neq x_t} Q_t(x_t', x_t)(s_\theta(x, t)_{x_t'} - \frac{p_{t|0}(x_t'|x_0)}{p_{t|0}(x_t|x_0)}\log s_\theta(x_t, t)_{x_t'} + C)dt$$

$$C = K\left(\frac{p_{t|0}(x_t'|x_0)}{p_{t|0}(x_t|x_0)}\right), \quad K(a) = a\log a - a,$$

Here $C$ is a constant irrelevant to the loss. The function $s_\theta(x_t, t)$ denotes a score network used to estimate the ratio $\frac{p_t(x_t')}{p_t(x_t)}$.

In practice, we often estimate $s_\theta(x, t)_{x_t'}$ using

$$s_\theta(x, t)_{x'} \approx \frac{p_t(x_t^1 \ldots x'^i_t \ldots x_t^d)}{p_t(x_t^1 \ldots x_t^i \ldots x_t^d)}$$

We denote $\tilde{Q}$ as the reverse transition matrix corresponding to $Q$. Then, the reverse process satisfies

$$\tilde{Q}_t(x_t, x_t') = \begin{cases} \frac{p_t(x_t')}{p_t(x_t)} Q_t(x_t', x_t) & \text{if } x_t' \neq x_t \\ -\sum_{k \neq x_t} \tilde{Q}_t(x_t, k) & \text{if } x_t' = x_t \end{cases}$$

Now we use Lemma 1 and calculate the three parts of the loss separately:

Using the fact that

$$\sum p_0(x'^i_t | x_t^{\text{unmasked}}) = 1$$

we obtain:

$$\int_0^T \mathbb{E}_{x_t \sim p_{t|0}(x_t|x_0)} \sum_{x_t' \neq x_t} Q_t(x_t', x_t) s_\theta(x, t)_{x_t'} dt = \int_0^T \mathbb{E}_{x_t \sim p_{t|0}(x_t|x_0)} \sum_{x_t^i = [M]} \beta_i f(t) \frac{\exp(-\beta_{x_t^i} \bar{f}(t))}{1 - \exp(-\beta_{x_t^i} \bar{f}(t))} dt$$

Using the properties of $-\frac{p_{t|0}(x_t'|x_0)}{p_{t|0}(x_t|x_0)}$ we obtain:

$$\int_0^T \mathbb{E}_{x_t \sim p_{t|0}(x_t|x_0)} \sum_{x_t^i \neq x_t} Q_t(x_t', x_t)\left(-\frac{p_{t|0}(x_t'|x_0)}{p_{t|0}(x_t|x_0)} \log s_\theta(x_t, t)_{x_t'}\right) dt$$

$$= \int_0^T \mathbb{E}_{x_t \sim p_{t|0}(x_t|x_0)} \sum_{x_t^i = [M], j \neq [M]} -f(t)\beta_i \mathbf{1}[x_0^i = j] \log s_\theta(x_t, t)_{x_t'} dt$$

Recalling that $K(a) = a\log a - a$ with $K(0) = 0$, we can safely ignore the terms $\mathbf{1}[x_0^i = j]$ and $j \neq [M]$.

$$\int_0^T \mathbb{E}_{x_t \sim p_{t|0}(x_t|x_0)} \sum_{x_t' \neq x_t} Q_t(x_t', x_t) K\left(\frac{p_{t|0}(x_t'|x_0)}{p_{t|0}(x_t|x_0)}\right)$$

$$= \int_0^T \mathbb{E}_{x_t \sim p_{t|0}(x_t|x_0)} \sum_{x_t^i = [M], j \neq [M]} \beta_i f(t) K\left(\frac{\exp(-\beta_{x_t^i} \bar{f}(t))}{1 - \exp(-\beta_{x_t^i}) \bar{f}(t)} \mathbf{1}[x_0^i = j]\right) dt$$

$$= \int_0^T \mathbb{E}_{x_t \sim p_{t|0}(x_t|x_0)} \sum_{x_t^i = [M]} \beta_i f(t) \frac{\exp(-\beta_{x_t^i} \bar{f}(t))}{1 - \exp(-\beta_{x_t^i}) \bar{f}(t)} \left(\log \frac{\exp(-\beta_{x_t^i} \bar{f}(t))}{1 - \exp(-\beta_{x_t^i}) \bar{f}(t)} - 1\right) dt$$

By adding them all together, especially combining the first and third term, we get

$$\int_0^T \mathbb{E}_{x_t \sim p_{t|0}(x_t|x_0)} \sum_{x_t^i = [M]} -\frac{\beta_i f(t) \exp(-\beta_i \bar{f}(t))}{1 - \exp(-\beta_i \bar{f}(t))} \log\left(\frac{\exp(-\beta_i \bar{f}(t))}{1 - \exp(-\beta_i \bar{f}(t))} q_\theta(x_0^i | x_t^{\text{unmasked}})\right) dt$$

Now we apply a change of variables. By defining

$$t_i = 1 - \exp\left(-\beta_{x_t^i}\,\bar{f}(t)\right), dt_i = \beta_{x_t^i} f(t) \exp\left(-\beta_{x_t^i}\,\bar{f}(t)\right)$$

the loss can be rewritten as:

$$\sum_i \int_0^1 \mathbb{E}_{x \sim p_{t|0}(x_t|x_0)}[-\frac{1}{t_i}\mathbf{1}[x_t^i = [\mathbf{M}]]\log(q_\theta(x_0^i|x_t^{\mathrm{unmasked}})]dt_i + C$$

Here, $x_{t_i}$ denotes the distribution in which $x^i$ is masked with probability $t_i$.

Recall that

$$t_i = 1 - \exp(-\beta_{x^i}\,\bar{f}(t)).$$

Suppose we have a reference value $\beta_{\mathrm{ref}}$ and define

$$t = 1 - \exp(-\beta_{\mathrm{ref}}\,\bar{f}(t)).$$

Then we can express $t_i$ as

$$t_i = 1 - (1-t)^{\frac{\beta_{x^i}}{\beta_{\mathrm{ref}}}}. \tag{21}$$

In practice, one can sample $t$ and calculate $t_i$ using the above formula. $\qquad\square$

## B  HYPERPARAMETERS

### B.1  TRAINING HYPERPARAMETERS

Our experiments are conducted with a learning rate of $1 \times 10^{-5}$ and a total of 20 training epochs. We apply gradient accumulation with 4 steps and used an effective per-device training batch size of 1. To reduce memory cost, we employ LoRA with rank $r = 128$, scaling factor $\alpha = 256$, and a dropout rate of 0.05. All hyperparameters, including model architecture, optimization, LoRA configuration, and training settings, are listed in Table 6.

| Category | Hyperparameter = Value |
|---|---|
| **Model Architecture** | d_model = 4096; n_layers = 32; n_heads = 32; n_kv_heads = 32; vocab_size = 126464; rope = True; rope_theta = 500000; pad_token_id = 126081; mask_token_id = 126336; layer_norm_type = rms; rms_norm_eps = 1e−5; alibi = False |
| **Optimization** | optimizer = adamw_torch; weight_decay = 0.1; learning_rate = 1e-5; lr_scheduler_type = linear; adam_beta1 = 0.9; adam_beta2 = 0.999; adam_epsilon = 1e-8; init_fn = mitchell; init_std = 0.02; max_grad_norm = 1 |
| **LoRA Configuration** | LoRA = `True`; $r$ = 128; lora_alpha = 256; lora_dropout = 0.05; target_modules = ["q_proj", "k_proj", "v_proj"] |
| **Training** | num_train_epochs = 20; per_device_train_batch_size = 1; gradient_accumulation_steps = 4; precision = bf16; max_sequence_length = 4096; seed = 42 |

Table 6: Hyperparameters used in training.

### B.2  EVALUATION HYPERPARAMETERS

In the evaluation, we set the block length to 32. For the two smaller datasets, Sudoku and Countdown, the generation length is set to 512; for the two larger datasets, MATH500 and GSM8K, the generation length is set to 256. The number of diffusion steps is always set to half of the generation length, and we use the low-confidence remasking method.

## C ALGORITHM PIPELINE

In Algorithm 1, we provide a detailed description of the implementation of WeFT.

---

**Algorithm 1:** WeFT: Weighted Entropy-driven Fine-Tuning

---

**Input:** model $\mathcal{M}$, input_ids $\mathcal{I}$, labels $l$, mask_token_id $m$, prompt lengths $L$
**Output:** Loss $\mathcal{L}$

**Function** EstimateRate($\mathcal{M}, \mathcal{I}, l, M, m$):
    // mask out the answer
    **for** $i \leftarrow 1$ **to** $|\mathcal{I}|$ **do**
        $\mathcal{I}'^i \leftarrow (1 - M^i) \cdot \mathcal{I}^i + M^i \cdot m$;
    $z \leftarrow \mathcal{M}(\mathcal{I}').\text{logits}$;
    // calculate entropy
    **for** $i \leftarrow 1$ **to** $|\mathcal{I}|$ **do**
        $H_i \leftarrow -\sum_j \text{softmax}(z_{ij}) \cdot \text{log\_softmax}(z_{ij})$;
        $\beta_i \leftarrow (1 - M^i)\sqrt{H_i}$;
    **return** $\beta$;

**Function** ComputeLoss($\mathcal{M}, \mathcal{I}, m, L$):
    $t \sim \mathcal{U}(0, 1)$;
    // mask the model's answer
    **for** $i \leftarrow 1$ **to** $|\mathcal{I}|$ **do**
        $M^i = \mathbf{1}_{\{i>L\}}$
    $\beta \leftarrow$ EstimateRate($\mathcal{M}, \mathcal{I}, M, m$);
    $\beta_{ref} \leftarrow \frac{\sum_i \beta_i \cdot M_i}{\sum_i M_i}$;
    // Compute the masking probability for each token
    **for** $i \leftarrow 1$ **to** $|\mathcal{I}|$ **do**
        $t_i \leftarrow 1 - (1 - t)^{\frac{\beta_i}{\beta_{ref}}}$;
        $M'^i \sim \text{Bernoulli}(t_i)$;
    $\mathcal{L} \leftarrow \frac{\sum_i \mathbf{1}_{\{M'^i=0\}} \frac{1}{t_i} \text{CrossEntropy}(z_i, l_i)}{\sum_i \mathbf{1}_{\{M'^i=0\}}}$;
    **return** $\mathcal{L}$;

---

## D USAGE OF LARGE LANGUAGE MODELS

In this paper, LLMs were used for polishing and improving phrases and sentences, but they did not participate in any aspects of idea conception, experimental design, theoretical proofs, figure creation, or paper structure design.

