# OpenReview forum: "WeFT: Weighted Entropy-driven Fine-Tuning for dLLMs"
_ICLR.cc/2026/Conference — ICLR 2026 Conference Withdrawn Submission_

### Official Review · Reviewer_boei · 2025-10-23

**Soundness:** 3
**Presentation:** 3
**Contribution:** 2
**Rating:** 4
**Confidence:** 4

**Summary:**

The paper introduces WeFT (Weighted Entropy-driven Fine-Tuning) for diffusion-based LLMs, addressing a key limitation of standard SFT: treating all tokens equally. In reasoning traces, some tokens carry far more informational weight; WeFT explicitly prioritizes them by estimating per-token uncertainty (entropy) in a probe pass and assigning higher mask rates and loss weights to the most uncertain positions. This diffusion-consistent, per-token weighting scheme concentrates learning where it matters most, yielding substantial gains on reasoning tasks with only modest additional compute.

**Strengths:**

**Originality:**

The paper moves from uniform token weighting to token adaptive weighting for diffusion LLMs, which is a novel SFT technique. It uses per token transition rates in the continuous time diffusion process, giving a diffusion consistent formulation rather than an ad hoc reweighting. It introduces a probe pass to estimate token entropy and converts that signal into both per token masking probabilities and per token loss weights, with square root entropy for stability.

**Quality:**

The paper is well structured, moving from motivation and CTDD derivation to an implementable algorithm and experiments. The theoretical analysis aligns with the modified forward process instead of a heuristic. The empirical evaluation on reasoning benchmarks includes informative ablations that support the causal explanation.

**Clarity:**

The writing is clear and easy to read. The core mechanisms, including the probe pass, per token rates, and weighting, are explained with useful intuition and equations. Figures and tables, as described, reinforce the presentation.

**Significance:**

- The method advances fine tuning practice for diffusion LLMs with a simple uncertainty aware objective that can be plugged into existing pipelines.

- It shows how to leverage the model’s own uncertainty to place training signal where it matters.

- The contribution is theoretically grounded and empirically validated and is likely to influence follow up work on uncertainty aware training.

**Weaknesses:**

*** Limited scope of application:** evaluated only on diffusion LLMs and reasoning tasks; How about scaling this method to LLMs

* Theory–practice gap (Eq. 13): using (\beta_i=\sqrt{H_i}) weakens diffusion matching; extend theory to (\beta_i=g(H_i)) with bounds, and include raw-entropy controls plus sensitivity sweeps. Not proper theoretical explanation of the empirical choice of the per-token weights.

* **Incomplete efficiency analysis:** only a single wall-time figure;  Additional analysis would be great. The  probing stage seems introducing  additional compute time. How do you propose reducing this computation cost.
 -

**Questions:**

see weaknesses

1. In the abstract, please clarify that “s1K” refers to specific datasets; name them (or briefly describe size/content) so the abstract is self-contained.

2. Is WeFT applicable to full-parameter fine-tuning (beyond LoRA)? If so, what are the compute and memory implications, and how do they scale with sequence length and batch size?

3. Can the method be adapted to other LLM families (e.g., autoregressive decoders, encoder–decoder models, multilingual/code LMs)? What changes, if any, are needed in the objective or masking schedule?

---

### Official Review · Reviewer_Asbj · 2025-10-29

**Soundness:** 3
**Presentation:** 3
**Contribution:** 2
**Rating:** 4
**Confidence:** 4

**Summary:**

This paper presents WeFT, a supervised fine-tuning method for diffusion language models. The central idea is to assign non-uniform training strength to different tokens according to their importance for generation, moving beyond standard SFT that uses a single masking rate and uniform loss weights. The method is evaluated on several categories of tasks and shows improvements over standard SFT. The paper includes several ablations and examines WeFT as an initializer for reinforcement learning.

**Strengths:**

1. This paper is well-written and carefully structured, with clarity maintained throughout the presentation, from the motivation to the detailed analysis of WeFT and its ablation studies. The content is presented in a manner that is easy to understand.

2. The paper provides a theoretical analysis to support the design of $t_i$, rooted in principles of diffusion-based language modeling. This analysis demonstrates rigor in linking theoretical derivations to practical implementation, enhancing the credibility of the proposed WeFT approach.

3. The experiments include multiple ablations, exploring practical choices for $\beta_i$ and comparing WeFT against standard SFT, WeFT (SW), and Dream. These ablations provide suggestive evidence for WeFT's effectiveness in practical contexts.

**Weaknesses:**

1. Concern about overclaiming. The abstract states that the method "achieves relative improvements of 39%, 64%, and 83% over standard SFT," while Table 1 shows small absolute gains in the averages; for example, moving from +0.6 (SFT) to +1.1 (WeFT) is summarized as 83% improvement.

2. Limited statistical robustness. Given the modest average gains in Table 1, multiple-seed runs with mean/variance and significance testing would strengthen the claims.

3. Limited generality of the RL result. RL cold-start is demonstrated only on Sudoku, followed by the statement that "models initialized with WeFT achieve higher rewards ... with a 49% relative performance gain." It remains unclear whether this holds on other tasks, which weakens the conclusion that WeFT is a better initializer.

4. The evidence for "higher entropy implies richer information" is relatively weak. Figure 2 provides qualitative word clouds and examples (e.g., "first/second" versus "all/such"), but this alone seems insufficient to support the claim. More evidence or analysis would strengthen it.

5. Typo: In line 327, "WeFTare" should be "WeFT are".

**Questions:**

Please see Weaknesses.

---

### Official Review · Reviewer_FjL4 · 2025-10-29

**Soundness:** 3
**Presentation:** 3
**Contribution:** 2
**Rating:** 4
**Confidence:** 4

**Summary:**

This paper addresses a key limitation in current diffusion-based language models, such as LLaDA and Dream, which treat all tokens equally during supervised fine-tuning (SFT). This approach is suboptimal, as certain tokens carry greater semantic weight within sentences and should accordingly be assigned higher importance during training. The authors propose WeFT (Weighted Fine-Tuning), an entropy-based method that first measures the model's confidence in predicting response tokens, then adjusts token weights based on these confidence scores. Experimental results demonstrate that when trained on s1k, s1k-1.1, and 3k samples extracted from Open-R1, WeFT achieves relative improvements of 39%, 64%, and 83% over standard SFT across four widely-used reasoning benchmarks (Sudoku, Countdown, GSM8K, and MATH-500).

**Strengths:**

- Diffusion language models constitute a rapidly developing research area, and improving the performance of their instruction-tuned versions is a relevant and timely research challenge.
- The paper demonstrates consistent gains of WeFT over standard SFT across four demanding reasoning benchmarks—Sudoku, Countdown, GSM8K, and MATH-500—suggesting the method’s broad applicability across diverse types of reasoning tasks.

**Weaknesses:**

- Formatting inconsistency: As per the ICLR 2026 submission guidelines, table captions should appear above tables. However, throughout the paper, captions are placed below tables, which violates the formatting requirements.
- Scalability concerns: Table 1 indicates that the average improvement of WeFT over SFT declines with larger training sets: s1k (0.9), s1k-1.1 (0.7), and 3k-OpenR1 (0.5). This diminishing trend raises questions about the method’s ability to scale effectively to larger datasets. Moreover, the relative performance gains appear limited.
- Limited experimental scope: Experiments are conducted only using LoRA fine-tuning. Including full-parameter fine-tuning would substantially strengthen the empirical validation. It remains unclear whether WeFT’s advantages persist under full fine-tuning settings.
- Computational cost vs. performance trade-off: WeFT introduces an additional forward pass per training step compared to SFT, yet the performance improvements are relatively limited. This invites doubt regarding the practical efficiency of the approach.

**Questions:**

- Following up on Weakness 2, could the authors provide a visualization illustrating how WeFT’s improvement over SFT varies with increasing training data size? Furthermore, does the approach continue to offer benefits when scaled to significantly larger datasets?
- Has WeFT been evaluated on benchmarks beyond those reported? A broader evaluation across diverse tasks would help better assess the generalizability of the method.
- Could the authors perform experiments applying WeFT directly to base models (rather than instruction-tuned ones) and compare against standard SFT? Such an ablation would offer valuable insights and strengthen the paper’s contribution.

---

### Official Review · Reviewer_Cyn9 · 2025-11-03

**Soundness:** 3
**Presentation:** 2
**Contribution:** 3
**Rating:** 4
**Confidence:** 5

**Summary:**

his paper introduces WeFT (Weighted Entropy-driven Fine-Tuning), a theoretically principled approach to supervised fine-tuning of
  diffusion language models (dLLMs) that addresses the limitation of uniform token treatment in standard SFT by assigning token-specific
  masking rates based on predictive entropy. The core contribution lies in deriving a weighted SFT loss formulation from continuous-time
  discrete diffusion theory by generalizing the transition rate matrix $Q$ to incorporate per-token masking rates $\beta_i$, which are
  computed as the square root of entropy $\beta_i = \sqrt{H(\text{softmax}(z_i))}$ from a preliminary forward pass with fully masked
  answers—this results in a loss where each token $i$ is masked with probability $t_i = 1 - (1-t)^{\beta_{x^i}/\beta_{\text{ref}}}$ and
  weighted by $1/t_i$. The experimental validation on LLaDA-8B-Instruct using small-scale datasets (s1K, s1K-1.1, and 3k samples from
  open-r1) demonstrates consistent but modest absolute improvements over standard SFT across four reasoning benchmarks (Sudoku, Countdown,
  GSM8K, MATH-500), with the reported 39-83% relative gains being computed against the SFT delta rather than baseline performance. While the
   theoretical derivation is sound and the entropy-based weighting is well-motivated through visualizations showing high-entropy tokens
  correspond to structurally important words ("first," "second"), the paper exhibits several limitations: (1) evaluation is restricted to a
  single 8B model architecture without exploring scalability to larger models where the benefits might be more pronounced, (2) the absolute
  performance gains are incremental (typically 1-3 percentage points), raising questions about practical significance, (3) the computational
   overhead of requiring two forward passes per training step (24% time increase) may become prohibitive at scale despite being
  characterized as "minimal," (4) ablation studies compare against only two alternative weighting schemes (NLL and Dream), leaving
  unexplored other potential metrics like gradient norms or attention scores, (5) the claim that entropy captures "reasoning and planning
  tokens" relies primarily on qualitative visualization of 100 high-frequency words rather than systematic linguistic or semantic analysis,
  and (6) training on only 1k-3k samples limits conclusions about data efficiency and convergence behavior on larger datasets typical of
  modern LLM training. The paper would benefit from multi-model validation, deeper analysis of what linguistic/semantic properties correlate
   with high entropy beyond the anecdotal examples provided, comparison with recent token-weighting methods from the autoregressive
  literature (e.g., Rho-1, DFT), and investigation of whether the benefits persist when scaling to larger models and datasets where the 24%
  training overhead becomes more costly—nevertheless, the work makes a solid theoretical contribution to diffusion-based language modeling
  and the consistent improvements across benchmarks and into RL training suggest the approach captures something meaningful about token
  importance in reasoning tasks.

**Strengths:**

- The paper removes a core limitation of prior dLLM SFT—uniform token masking—by introducing a per-token diffusion rate $\beta_i$ and a principled mapping $t\mapsto t_i=1-(1-t)^{\beta_i/\beta_{\mathrm{ref}}}$, yielding a weighted objective with tokenwise weights $1/t_i$. This is a creative synthesis of continuous-time discrete diffusion theory, DSE-style training, and uncertainty-aware token selection via $\sqrt{\text{entropy}}$, turning a heuristic idea ("focus on hard tokens") into a diffusion-consistent formulation rather than an ad-hoc reweighting.

- The work is technically careful: it presents the modified generator matrix $Q$, derives the new loss with lemmas and a theorem, and supplies an efficient two-pass estimator for $\beta_i$ that adds only one extra forward pass. The empirical study is solid for a methods paper: gains are demonstrated across four reasoning benchmarks and three training sets, reinforced by ablations (NLL-weighting vs. entropy, simple token-wise weights vs. the proposed diffusion-consistent loss), an RL cold-start transfer showing benefits persist beyond SFT, and a runtime analysis (~24% overhead). The inclusion of Algorithm-level pseudocode and complete hyperparameters (incl. LoRA settings) increases reproducibility.

- The exposition is easy to follow: the paper motivates token importance, anchors it in predictive entropy, and walks the reader from diffusion preliminaries to the weighted objective with consistent notation and numbered equations; Figures 1–2 and the algorithm box precisely illustrate the training pipeline, and tables report both absolute metrics and relative deltas, helping readers gauge effect sizes.

- If diffusion LMs continue to mature, a drop-in, theory-aligned SFT objective that consistently improves reasoning accuracy and even accelerates downstream RL learning has broad impact: it directly benefits data-limited instruction tuning, makes dLLMs more controllable by emphasizing structurally pivotal tokens, and suggests a general recipe—uncertainty-aware token scheduling under a principled dynamics model—that could transfer to other discrete diffusion or even AR setups; the simplicity (one extra pass) further lowers the barrier to adoption for practitioners.

**Weaknesses:**

While I appreciate the theoretical analysis , I have some concerns about the mathematical foundations and claims that need to be addressed.

## Major Theoretical Problems

Let me start with what I consider the two most fundamental issues, both of which undermine the theoretical justification for the method.

### The derivation assumes a fixed Q, but the algorithm uses an adaptive one

* **In §2 (Preliminaries)** you assume $Q_t=f(t)Q$ with a **constant** $Q$ (Eq. 3, 5, 7).
* **In §3.2**, you generalize $Q$ to include per‑token rates $\beta$ (Eq. 9), but this still reads as a *fixed* matrix for the process.
* **In §3.3/Algorithm 1**, $\beta_i$ is set **per example, per step, from the current model's logits** via $\sqrt{H(\text{softmax}(z_i))}$. That makes $Q$ **depend on both data and parameters $\theta$**.

This is a problem because the Kolmogorov equations and the change‑of‑variables steps used to obtain Eq. (10)/(21) assume a Markov process with a generator that's independent of the evolving state and model parameters. Once $Q$ depends on $x$ and $\theta$, those identities no longer hold as stated. At minimum, the proof needs to be redone for a **state‑dependent generator**—or alternatively, the method should be repositioned as a **sampling/weighting heuristic** rather than a theorem‑backed estimator.

There are two potential paths forward here. You could freeze $\beta_i$ using a reference model or a data‑dependent but **parameter‑independent** proxy, which would keep $Q$ fixed during optimization and make your theorem applicable. Or you could acknowledge that the theorem only covers the fixed‑$Q$ case and present the entropy‑driven masking as a heuristic variance‑control scheme. I'd personally prefer seeing the latter with an honest discussion of why it works empirically.

### The objective doesn't actually upweight high‑entropy tokens

This is subtle but important. Your loss (Eq. 10) is

$$
L=\sum_i \mathbb{E}_{t_i}\big[ \mathbf{1}[x^i_{t_i}=\mathbf{M}] \, \tfrac{1}{t_i}\, \log p_\theta(x_0^i \mid x_t) \big].
$$

(There's also a missing minus sign here—more on that below.)

Let me walk through why this doesn't prioritize high-entropy tokens the way the paper claims. If we let $\ell_i=-\log p_\theta(x_0^i\mid x_t)$ and note that $\mathbf{1}[x^i_{t_i}=\mathbf{M}]\sim \text{Bernoulli}(t_i)$, then

$$
\mathbb{E}\big[\mathbf{1}[x^i_{t_i}=\mathbf{M}]\tfrac{1}{t_i}\,\ell_i\big]
= \ell_i.
$$

So the **expected** contribution per token is unchanged—no token is prioritized in expectation, regardless of $t_i$. What you actually get is **more frequent sampling** of high‑entropy tokens, but the inverse‑probability weighting ($1/t_i$) exactly cancels out the frequency change. The benefit here is plausibly **variance reduction**, not reweighting as claimed throughout the paper (Abstract, §3.1, §3.3).

It would make more sense to reframe this as **importance sampling** or **variance control** rather than "prioritization." Alternatively, if you genuinely want to upweight difficult tokens, you'd need to either remove the $1/t_i$ term (accepting bias) or use a partial correction like $1/t_i^\alpha$ with $\alpha<1$ and then analyze the bias–variance tradeoff explicitly.

## Mathematical Issues in the Derivation

Beyond the conceptual problems, there are several concrete mathematical errors that need fixing.

**Sign errors and missing negatives:** Equations (1) and (10) write $\log(x_0^i|x_t)$ as part of the **loss**, but a loss should use $-\log p_\theta(\cdot)$ or an explicit cross-entropy term. Algorithm 1 does use `CrossEntropy`, which implies the negative sign, so there's an inconsistency here.

**Small-o notation:** Equation (2) uses $o(t)$ where it should be $o(\Delta t)$.

**Lemma 1:** This one really caught my attention. In Equation (18), the numerator and denominator are identical:

$$
\frac{p_t(x_t^1,\dots,x_t^i,\dots,x_t^d)}{p_t(x_t^1,\dots,x_t^i,\dots,x_t^d)}
= \frac{\exp(-\beta_{x_t^i}\bar f(t))}{1-\exp(-\beta_{x_t^i}\bar f(t))}\
 p_0(x_t'^i\mid x_t^{\text{unmasked}}). \tag{18}
$$

The LHS is just 1. I think you meant to write a ratio of two *different* sequences—maybe something like $\frac{p_t(x_t^{(i=[M])})}{p_t(x_t^{(i=x'_t)})}$, or $p_t(\text{masked at }i)/p_t(\text{unmasked with token }x'_t)$. Equation (19) has the same issue. As currently written, Lemma 1 is invalid.

**Change-of-variables in Appendix A:** You write "define $t_i=1-\exp(-\beta_{x_t^i}\bar f(t))$ and $dt_i=\beta_{x_t^i} f(t)\exp(-\beta_{x_t^i}\bar f(t))$," which is differentiation w.r.t. the time variable. But then your integral bounds are over $t_i\in[0,1]$ without a clear statement of how the measure changes. Combined with the state-dependent $Q$ issue from earlier, this section needs careful rewriting.

**Notation inconsistencies:** There's a lot of notation drift—switches between $\beta_{xi}$, $\beta_{x^i}$, and $\beta_i$

## Problems with Algorithm 1

The pseudocode has several concrete bugs that need fixing before the method can be correctly implemented. These aren't conceptual issues—they're implementation errors that would prevent the algorithm from working as intended.

**You're training on the wrong tokens.** The loss computation uses

$$
\mathcal{L} \leftarrow \frac{\sum_i \mathbf{1}_{\{M'^i=\mathbf{0}\}} \frac{1}{t_i}\,\text{CE}(z_i,l_i)}
{\sum_i \mathbf{1}_{\{M'^i=\mathbf{0}\}}}.
$$

But Eq. (10) says you should train on $\mathbf{1}[x_{t_i}^i=\mathbf{M}]$—the **masked** tokens. Your code trains on unmasked tokens (`M'^i=0`), which inverts the training signal.

**The second forward pass is missing.** After you sample $M'$ with $\text{Bernoulli}(t_i)$, you need to build the partially masked input and run the model again to get logits $z$ conditioned on that specific masking pattern. In the pseudocode, `z` only comes from the first pass (where all answers are masked), not from the actual training masking pattern.

**$\beta_i$ is zeroed for answer tokens.** In `EstimateRate`, you set $\beta_i \leftarrow (1-M^i)\sqrt{H_i}$. But earlier, $M^i=1$ denotes the answer tokens (the ones you're masking). This assignment zeros out $\beta_i$ for exactly the tokens you want to weight. It should be $\beta_i \leftarrow M^i\sqrt{H_i}$.

**Undefined variables in the loss.** The `z_i` used in the CE is never properly defined (see missing second forward pass above). Also, `l` is passed to `EstimateRate` but never actually used.

**Unexpected normalization.** Equation (10) doesn't divide by the number of masked tokens, but your algorithm divides by $\sum \mathbf{1}[M'^i=0]$. If you want an unbiased estimator of the sum, you shouldn't include this normalization—or if you want to redefine the objective as an average, the weighting needs to be adjusted accordingly.

## Experimental Claims Don't Match the Results

This is perhaps the most visible problem in the paper. When I read the abstract claiming "39%, 64%, and 83% relative improvements," I immediately went to Table 1 to see these impressive gains—but the numbers tell a very different story.

### The claimed improvements vs. what's actually in the tables

The abstract and §4.2 state: "WeFT delivers substantial gains… **39%, 64%, and 83%** relative improvements over SFT…"

But looking at Table 1's averages:
- s1K: SFT gets **34.9**, WeFT gets **35.8** → that's a **+2.6%** relative improvement
- s1K‑1.1: **33.7** → **34.4** → **+2.1%** relative
- openr1‑3k: **33.2** → **33.7** → **+1.5%** relative

These are nowhere near 39/64/83%. Even looking at individual tasks, WeFT sometimes **underperforms** SFT (e.g., on s1K‑1.1, Countdown is 20.3 vs. 21.5; GSM8K is 78.1 vs. 78.4). So the claim that "WeFT consistently achieves higher accuracy than SFT" isn't supported by Table 1.

I'm genuinely confused about where the 39/64/83% numbers came from. If they refer to some other metric (perhaps error reduction on Sudoku specifically? or improvement relative to SFT's delta from baseline?), this needs to be stated explicitly and calculated transparently in the text.

**"SFT improves performance on all tasks"** — §4.2 states: "Compared with the base… both SFT and WeFT improve performance on all tasks." But on Sudoku (s1K), SFT actually **drops** from 5.5 to 4.6.

**Questions:**

Please see above sectino.

---

### Note · Authors · 2025-11-12

**Comment:**

Thanks for the detailed review. They are very helpful for improving our work. We decided to withdraw the paper to further improve its quality.

**Withdrawal Confirmation:**

I have read and agree with the venue's withdrawal policy on behalf of myself and my co-authors.